

# Identification of a novel potassium channel (GiK) as a potential drug target in *Giardia lamblia*: Computational descriptions of binding sites

Lissethe Palomo-Ligas[1], Filiberto Gutiérrez-Gutiérrez[2],
Verónica Yadira Ochoa-Maganda[1], Rafael Cortés-Zárate[3],
Claudia Lisette Charles-Niño[3] and Araceli Castillo-Romero[3]

[1] Departamento de Fisiología, Centro Universitario de Ciencias de la Salud, Universidad de Guadalajara, Guadalajara, Jalisco, Mexico
[2] Departamento de Química, Centro Universitario de Ciencias Exactas e Ingenierías, Universidad de Guadalajara, Guadalajara, Jalisco, Mexico
[3] Departamento de Microbiología y Patología, Centro Universitario de Ciencias de la Salud, Universidad de Guadalajara, Guadalajara, Jalisco, Mexico

Corresponding author
Araceli Castillo-Romero,
araceli.castillo@cucs.udg.mx

## ABSTRACT

**Background:** The protozoan *Giardia lamblia* is the causal agent of giardiasis, one of the main diarrheal infections worldwide. Drug resistance to common antigiardial agents and incidence of treatment failures have increased in recent years. Therefore, the search for new molecular targets for drugs against *Giardia* infection is essential. In protozoa, ionic channels have roles in their life cycle, growth, and stress response. Thus, they are promising targets for drug design. The strategy of ligand-protein docking has demonstrated a great potential in the discovery of new targets and structure-based drug design studies.

**Methods:** In this work, we identify and characterize a new potassium channel, GiK, in the genome of *Giardia lamblia*. Characterization was performed *in silico*. Because its crystallographic structure remains unresolved, homology modeling was used to construct the three-dimensional model for the pore domain of GiK. The docking virtual screening approach was employed to determine whether GiK is a good target for potassium channel blockers.

**Results:** The GiK sequence showed 24–50% identity and 50–90% positivity with 21 different types of potassium channels. The quality assessment and validation parameters indicated the reliability of the modeled structure of GiK. We identified 110 potassium channel blockers exhibiting high affinity toward GiK. A total of 39 of these drugs bind in three specific regions.

**Discussion:** The GiK pore signature sequence is related to the small conductance calcium-activated potassium channels (SKCa). The predicted binding of 110 potassium blockers to GiK makes this protein an attractive target for biological testing to evaluate its role in the life cycle of *Giardia lamblia* and potential candidate for the design of novel antigiardial drugs.

## INTRODUCTION

*Giardia lamblia* is the causal agent of giardiasis, a prolonged diarrheal disease. The standard compounds used against *Giardia lamblia* are 5-nitroimidazoles. However, these compounds present side effects associated with residual toxicity in the host. Dose-dependent side effects include leukopenia, headache, vertigo, nausea, insomnia, irritability, metallic taste, and CNS toxicity (*Ansell et al., 2015*; *Escobedo & Cimerman, 2007*; *Tejman-Yarden & Eckmann, 2011*; *Watkins & Eckmann, 2014*). In addition, reports of resistant strains and nitroimidazole-refractory disease are of considerable concern. Reduced efficacy has been described even with higher drug doses (*Carter et al., 2018*; *Leitsch, 2015*). For these reasons, there is a significant need for identification of new anti-*Giardia* drugs and drug targets. Ionic channels are pore-forming proteins that allow the passage of specific ions across the membrane, regulating different physiological processes (*Subramanyam & Colecraft, 2015*). Because of their biophysical behavior and participation in different human pathologies, ionic channels are attractive targets for drug design (*Bagal et al., 2013*). Potassium channels are the most diverse and ubiquitous group of ion channels. They are divided into four main families on the basis of their biophysical and structural properties: voltage-gated $K^+$ channels, calcium-activated $K^+$ channels ($K_{Ca}$), inward-rectifier $K^+$ channels and two-pore-domain $K^+$ channels ($K_{2P}$) (*Wulff, Castle & Pardo, 2009*). In both electrically excitable and non-excitable cells, potassium channels regulate multiple cellular functions including cell volume, proliferation, differentiation, and motility (*Grunnet et al., 2002*; *Pchelintseva & Djamgoz, 2018*; *Schwab et al., 2008*; *Urrego et al., 2014*).

Recently, several studies have reported identification and characterization of $K^+$ channels in pathogenic protozoa. In *Plasmodium falciparum* and *Trypanosoma cruzi*, these channels are expressed in different stages of the parasite life cycle. They are essential for growth and play a significant role in parasite response to environmental stresses (*Ellekvist et al., 2004*; *Jimenez & Docampo, 2012*; *Waller et al., 2008*). A heterodimeric $Ca^{2+}$-activated potassium channel was identified in *Trypanosoma brucei*. This identification was accomplished by profile searches of the predicted parasite proteome against the conserved loop of cation channels. The channel identified was found to be essential for the bloodstream form parasites (*Steinmann et al., 2015*). The National Center for Advancing Translational Sciences Small Molecule Repository was screened. In this screening, fluticasone propionate was identified as a potential good inhibitor of *T. brucei* potassium channels. Experiments confirmed fluticasone propionate as a candidate drug targeting *T. brucei* (IC50 of 0.6 μM) (*Schmidt et al., 2018*). Biaguini and coworkers showed that $K^+$ causes an important depolarization of the membrane in *Giardia lamblia* (*Biagini et al., 2000*). Results of others studies, report that $K^+$ plays an important role as an osmolyte regulating *Giardia* cell volume (*Maroulis, Schofield & Edwards, 2000*). *Xenopus* oocytes were injected with mRNA isolated from trophozoites of *Giardia lamblia*, subsequent electrophysiology experiments revealed potassium currents (*Ponce, Jimenez-Cardoso & Eligio-Garcia, 2013*). By genome analysis and a bioinformatic approach, Prole and Marrion identified a putative potassium channel in

*Giardia lamblia* assemblage E (*Prole & Marrion, 2012*). However, the structural characterization of ionic channels in this protozoan is limited. Consequently, the potential of these channels to serve as a drug targets is poorly understood.

In recent years, *in silico* strategies have been used frequently to estimate protein function, for the discovery of new target molecules and for structure-based drug design studies (*Chen & Chen, 2008*). This work describes computational approaches to determine structural biology of a putative *Giardia* potassium channel, GiK. Further, this work evaluates the potential of this channel to serve as a novel target. A closed-state pore domain of GiK homology model was constructed. This construction was accomplished using a high conductance calcium-activated potassium channel from *Aplysia californica* (PDB ID: 5TJI) as a template. Our docking and virtual screening approach identified 110 potassium channel blockers exhibiting high free energy of binding to GiK, 39 of these drugs bind in the pore region of the channel. The drugs interact mainly with sites in three specific regions: S5, S2–S4 and C-terminal. These findings support the conclusion that this protein is an attractive target for biological testing to reveal its role in the life cycle of *Giardia lamblia* and a potential candidate for the design of novel antigiardial drugs.

## MATERIALS AND METHODS

### *In silico* putative potassium channel identification in *Giardia lamblia*

To identify homologous sequences in *Giardia lamblia*, 51 potassium channel sequences from genomes of different species, deposited in the NCBI protein database (http://www.ncbi.nlm.nih.gov/protein), were compared by BLAST algorithm with the *Giardia* genome database (http://giardiadb.org/giardiadb/).

The amino acid composition, physicochemical properties, solvation and protein binding sites of the resulting sequence (GiK) (Accession number XP_001709490) were analyzed using PROTPARAM ((http://expasy.org/tools/) and PredictProtein (*Yachdav et al., 2014*). We applied Predictor of Natural Disordered Regions (*Obradovic et al., 2003*) to predict disorder regions. Highly conserved residues were identified by consensus results of NCBI Conserved domains (*Marchler-Bauer et al., 2017*), Motif Search (http://www.genome.jp/tools/motif/), InterProScan tool (*Jones et al., 2014*), Block Searcher (*Henikoff & Henikoff, 1994*), and ExPASy PROSITE (*Sigrist et al., 2013*). Consensus results of the Constrained Consensus Topology prediction server (*Tusnady & Simon, 1998*, *2001*) and PredictProtein (*Yachdav et al., 2014*) servers were used for the prediction of transmembrane domains.

### Prediction of the potassium blockers binding sites on GiK
#### *Homology model and refinement*

The crystal structure of GiK is not available. Therefore, three-dimensional (3D) models of the pore region (1–500 aa) were produced using I-TASSER (Iterative Threading ASSEmbly Refinement) (*Roy, Kucukural & Zhang, 2010*; *Yang et al., 2015*; *Zhang, 2008*), RaptorX (*Ma et al., 2012*, *2013*; *Peng & Xu, 2010*), Phyre2 (Protein Homology/analogY Recognition Engine V 2.0) (*Kelley et al., 2015*), SWISS-MODEL (*Arnold et al., 2006*; *Biasini et al., 2014*; *Bordoli et al., 2008*), and Modeller 9.18 (*Fiser, Do & Sali, 2000*;

*Martí-Renom et al., 2000*; *Šali & Blundell, 1993*; *Webb & Sali, 2014*). First, we searched the PDB (*Berman et al., 2007*) for known protein structures using the GiK sequence as query. We also searched for suitable templates in the SWISS-MODEL Template library. Next, a multiple alignment of the GiK sequence (UniProtKB accession: A8B451) to the main template structures was calculated, by MultAlin software (*Corpet, 1988*). Optimization of the hydrogen bonding network and the atomic level energy minimization of the 3D-GiK models generated were performed using the What If Web Interface (*Chinea et al., 1995*) and the 3D Refine protein structure refinement server (*Bhattacharya & Cheng, 2013*; *Bhattacharya et al., 2016*). The global structural quality of predicted models was validated by RAMPAGE (Ramachandran Plot Analysis) (*Lovell et al., 2003*), QMEAN (Qualitative Model Energy Analysis) (*Benkert, Tosatto & Schomburg, 2008*), Verify 3D (*Bowie, Luthy & Eisenberg, 1991*; *Luthy, Bowie & Eisenberg, 1992*), ERRAT (*Colovos & Yeates, 1993*) and ProSA-web (*Wiederstein & Sippl, 2007*). The 3D-GiK model with the best scoring was selected for refinement using UCSF CHIMERA v1.11.1 (*Pettersen et al., 2004*). We used 100 steps of conjugate gradient minimization. The QMEANBrane tool was used to assess the local quality of the 3D-GiK membrane protein model (*Studer, Biasini & Schwede, 2014*). To confirm the quality of the models, we compare the 13 resulting 3D models with the corresponding experimental structure using the root mean square deviation (RMSD). TM-align was used to determinate the backbone C$\alpha$ coordinates of the given protein structures. The results of the predicted models with C$\alpha$-RMSD are expressed in Å. The monomer was built by alignment with template 5TJI. Tetrameric assemblage was obtained by the Maestro 2017-1 software with four holo forms monomers of 5TJIs, avoiding overlapping of monomers (Schrödinger, LLC, New York, NY, USA).

## Molecular docking evaluation

Numerous structures of potassium blockers have been reported. To identify potential drug binding sites on the GiK protein, we selected 290 potassium blockers from the Drug bank (www.drugbank.ca), Sigma profile (www.sigmaaldrich.com) and Zinc (http://zinc.docking.org) (*Irwin et al., 2012*) databases. Prior to docking, all structures were energy minimized using Maestro 2017-1 (Schrödinger, LLC, New York, NY, USA).

The docking simulations were carried out using AutoDock Vina software, employing a Lamarckian genetic algorithm (*Trott & Olson, 2010*), with a grid box of 126 Å$^3$ and nine binding modes. The complexes and poses between 3D-GiK and potassium blockers were analyzed using Maestro 2017-1 (Schrödinger, LLC, New York, NY, USA). The results are reported as binding energy of ligand and protein in kcal/mol.

## RESULTS

### Identification and characterization of the putative potassium channel GiK

We performed BLAST searches of the *Giardia* genome database. We used the whole sequence of 51 potassium channels genomic sequences of different species as queries (Table S1). The uncharacterized protein GL50803_101194, GiK (GenBank Accession: XP_001709490),

**Table 1  Sequences producing significant alignments with GiK by BLAST.**

| Accession number | Organism | Type of channel | Score | E. value | Identities | Positives |
|---|---|---|---|---|---|---|
| BAN90095.1 | *Aeropyrum camini* | Kv | 32 | 0.33 | 15/49 (31%) | 29/49 (59%) |
| AEE68730.1 | *Bordetella pertussis* | Kv | 32 | 0.51 | 15/43 (35%) | 25/43 (58%) |
| WP_012338231.1 | *Burkholderia cenocepacia* | Kv | 33 | 0.20 | 16/39 (41%) | 25/39 (64%) |
| YP_002407586.1 | *Escherichia coli* | Kv | 28.9 | 8.6 | 18/66 (27%) | 36/66 (55%) |
| WP_024212520.1 | *Escherichia spp* | Multispecies Kv | 28.9 | 9.8 | 18/66 (27%) | 36/66 (55%) |
| AAP94028.1 | *Gallus gallus* | Kv1.3 | 34.7 | 0.27 | 20/58 (34%) | 32/58 (55%) |
| WP_011570442.1 | *Haloquadratum walsby* | Ion channel | 33 | 0.18 | 10/20 (50%) | 18/20 (90%) |
| AAA61276.1 | *Homo sapiens* | Kv | 35 | 0.24 | 16/43 (37%) | 25/43 (58%) |
| NP_002223.3 | *Homo sapiens* | Kv1.3 | 35.0 | 0.17 | 20/58 (34%) | 32/58 (55%) |
| CDS30290.2 | *Hymenolepis microstoma* | Kv | 32.7 | 2.4 | 22/80 (28%) | 40/80 (50%) |
| AEO96823.2 | *Lateolabrax japonicus* | Kv1.3 | 33.1 | 0.56 | 18/45 (40%) | 26/45 (58%) |
| CCQ21618.1 | *Listeria monocytogenes* | Kv | 36 | 0.012 | 14/41 (34%) | 27/41 (66%) |
| NP_001245037.1 | *Macaca mulatta* | Kv1.3 | 35.0 | 0.18 | 20/58 (34%) | 32/58 (55%) |
| NP_032444.2 | *Mus musculus* | Kv1.3 | 33.5 | 0.47 | 14/38 (37%) | 24/38 (63%) |
| XP_007383667.1 | *Punctularia strigosozonata* | Kv | 44 | 5e-04 | 33/105 (31%) | 53/105 (50%) |
| WP_006887331.1 | *Rothia aeria* | Kv | 36.6 | 0.032 | 17/67 (25%) | 34/67 (51%) |
| NP_707157.2 | *Shigella flexneri 2a str. 301* | Kv | 29 | 5.3 | 18/66 (27%) | 36/66 (55%) |
| CAA56175.1 | *Solanum tuberosum* | Kir | 32.0 | 1.9 | 18/67 (27%) | 34/67 (51%) |
| NP_631700.1 | *Streptomyces coelicolor* | Kv | 30 | 0.54 | 9/34 (26%) | 24/34 (71%) |
| CDW52461.1 | *Trichuris trichiura* | Kv | 31.6 | 1.9 | 12/49 (24%) | 27/49 (55%) |
| AUI87359.1 | *Vibrio azureus* | Kv | 31.6 | 1.3 | 27/91 (30%) | 46/91 (51%) |

**Table 2  Physicochemical characterization of GiK by Protparam.**

| | |
|---|---|
| Number of amino acids | 1,416 |
| Molecular weight | 25,811.2 |
| Instability index | 45.47 |
| Aliphatic index | 93.28 |
| Grand average of hydropathicity (Gravy) | −0.053 |
| Isoelectric point | 8.18 |
| Ext. Coeficiente | 141,880 |

showed 24–50% identity and 50–90% positivity with 21 different types of voltage-gated potassium channels (Table 1). Physicochemical properties were obtained (Table 2). These properties enabled establishment of GiK molecular weight, stability index, isoelectric point, aliphatic index, and Grand Average of Hydropathicity (GRAVY) of GiK.

The instability index indicates that GiK might be unstable in nature (instability index > 40). The aliphatic index, a factor in protein thermal stability, is related to the mole fraction of Ala, Ile, Leu, and Val in the protein. The aliphatic index of GiK 93.28 indicates a thermally stable protein that contains high amount of hydrophobic amino acids (Fig. S1). The negative value of GRAVY indicates that GiK is a hydrophilic protein (*Wilkins et al., 1999*). The prediction of disordered regions in GiK suggests that this protein has 11 intrinsically

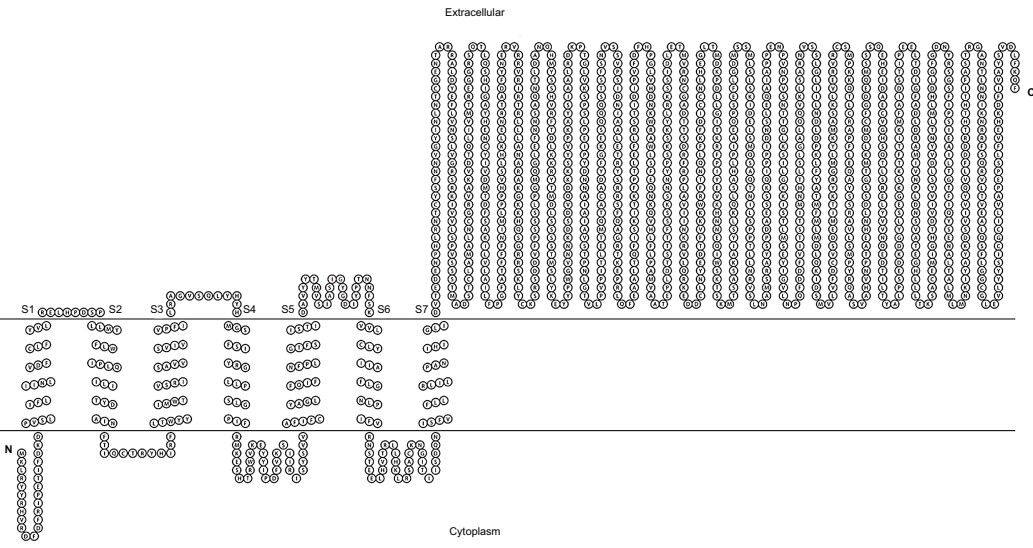

**Figure 1 Transmembrane structure of GiK.** It contains seven transmembrane segments (S1–S7), the P-loops between S5 and S6 form the pore domain. The selectivity filter is in gray.

disordered regions (IDRs) that could be involved in important *Giardia* functions (Fig. S2). The membrane topology and the analysis of the main features of K⁺ channels show that GiK is a membrane protein that possesses seven helical transmembrane (HTM) regions. Further, evidence shows a highly conserved pore-loop sequence that determines K⁺ channel selectivity (Fig. 1). According to databases of protein signatures, GiK contains: a domain related to ionic channels, Ion_trans_2 domain; domains related to voltage-gated potassium channels, 215625 and 236711; one domain associated with signal transduction, 227696; two fingerprints of potassium channel, 2POREKCHANEL and KCHANNEL; and one fingerprint related with EAG/ELK/ERG channels (EAGCHANLFMLY). These results suggest that this protein is a potassium channel (Fig. 2; Table 3).

The pore-forming domain is highly conserved in all types of K⁺ channels. An alignment revealed that all sequences that showed homology with GiK present the pore signature sequence S/TXGXGX. GiK has the residues SIASIGYGD, similar to TFLSIGYG, which are present in small conductance calcium-activated potassium channels (SKCa) (*Shin et al., 2005*) (Fig. 3). Finally, using PredictProtein server (*Yachdav et al., 2014*), we predicted GiK has potassium channel activity with 36% reliability.

## Modeling and structure quality of GiK protein

The prediction of the 3D-GiK structure was done by homology modeling. The search for a structural template for GiK protein revealed identity with four resolved protein structures. Two structures were the open and closed state of a high conductance calcium-activated potassium channel from *A. californica* (PDB ID: 5TJ6, open state, and 5TJI, closed state), with 23% sequence identity. The other two structures were the open and closed state of a potassium channel subfamily T member one from *Gallus gallus* (PDB: 5U70, open state, and 5U76, closed state) with 19% sequence identity (Fig. S3).

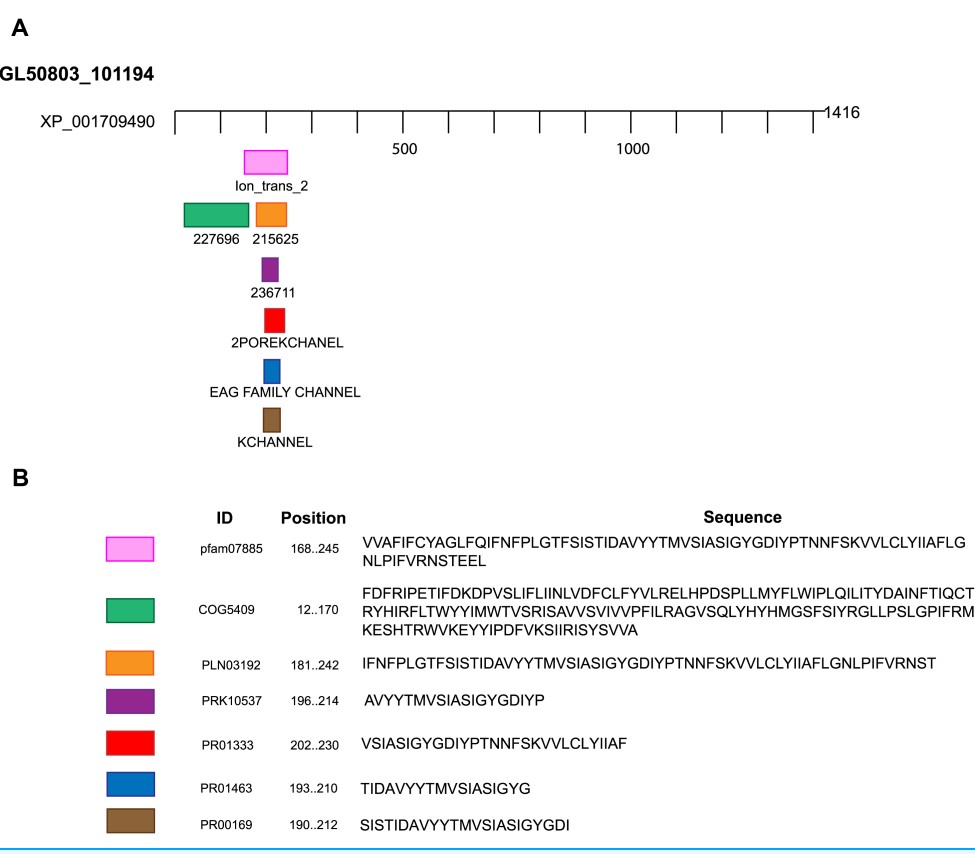

**Figure 2 Domains and motifs related to potassium channels.** GiK presents domains related to different subtypes of potassium channels. (A) Schematic representation. (B) Accession number and description of the sequences.

**Table 3 Prediction of highly conserved residues from GiK.**

| Domain or motif | Description | Accession number | Position (E value) | Server |
|---|---|---|---|---|
| Ion_trans_2 | Ionic channel. This family includes the two membrane helix type ion channels found in bacteria. | pfam07885 | 168–245 (1.35e-08) | NCBI Conserved domains, Motif search, InterProScan tool |
| 227696 | EXS domain-containing protein (Signal transduction mechanisms). | COG5409 | 12–170 (0.44) | ExPASy PROSITE, Motif search |
| 215625 | Voltage-dependent potassium channel; Provisional. | PLN03192 | 181–242 (0.14) | ExPASy PROSITE, Motif search |
| 236711 | Voltage-gated potassium channel; Provisiona.l | PRK10537 | 196–214 (0.70) | ExPASy PROSITE, Motif search |
| 2POREKCHANEL | Potassium channel domain. | PR01333 | 202–230 (0.00032) | Block searcher |
| EAGCHANLFMLY | EAG/ELK/ERG potassium channel family signature. | PR01463 | 193–210 (0.029) | Block searcher |
| KCHANNEL | Potassium channel signature. | PR00169 | 190–212 (0.1) | Block searcher |

Model construction was performed using five homology modeling servers: I-TASSER, RaptorX, Phyre2, Swiss model, and Modeller 9.18. Using the four templates, a modeling protocol was constructed for each program. The final dataset includes
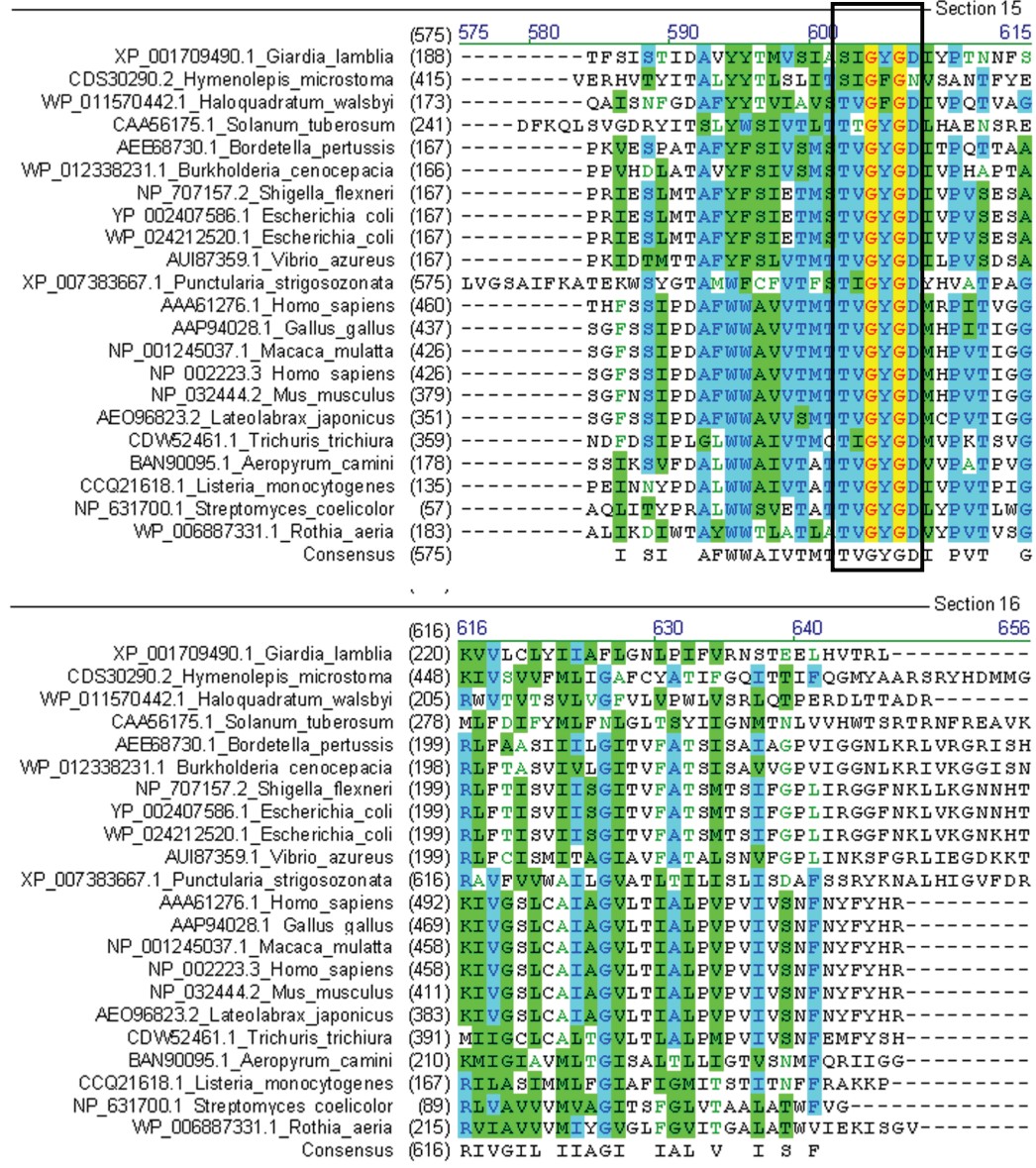

**Figure 3 Multiple sequence alignment of GiK with voltage-gated potassium channels.** The signature sequence T/SXGXGX of the selectivity filter is present in all classes of potassium channels (black square).

13 3D-GiK models covering a wide range of quality. The global quality of each theoretical model was validated by the Ramachandran plot analysis, QMEAN score, Z score, ERRATscore, and Verify 3D. Modeller 9.18 program produced the best 3D-GiK model, using the sequence 5TJI as template (Table 4). Figures 4A–4C show the resulting ratio of Z-score and the QMEAN score obtained for GiK. The z-score value, −5.07, is in the range of native conformations. This can be seen clearly when the score is compared to the scores of other experimentally determined protein structures with the same number of residues. Further, the QMEAN4 score is in the range of a good experimental structure (0.296). Additionally, the Ramachandran plot analysis confirms that this model is

**Table 4 Validation scores from RAMPAGE, QMEAN, ProSA-web, ERRAT, and Verify 3D of the constructed models.**

| Software | Template (PDB ID) | Ramachandran (%) | QMEAN score | Z-score | ERRAT score | Verify 3D | Residues | RMSD (Å) |
|---|---|---|---|---|---|---|---|---|
| **Modeller** | 5TJ6 | 90.4 | 0.141 | −7.56 | 44.26 | 26.28 | 500 | 4.28 |
| | 5U70 | 90.0 | 0.094 | −8.09 | 39.62 | 14.06 | 500 | 5.05 |
| | 5TJI | 94.2 | 0.296 | −5.07 | 69.24 | 35.60 | 500 | 3.90 |
| | 5U76 | 88.4 | 0.023 | −9.22 | 34.97 | 26.28 | 500 | 4.46 |
| **Raptorx** | 5TJ6 | 89.8 | 0.191 | −6.92 | 56.64 | 20.60 | 500 | 4.85 |
| **I-tasser** | 5TJ6 | 72.9 | 0.101 | −8.78 | 86.58 | 38.80 | 500 | 3.97 |
| | 5U70 | 69.6 | 0.089 | −9.12 | 81.91 | 44.60 | 500 | 5.01 |
| **Swiss model** | 5TJ6 | 89.8 | 0.205 | −6.21 | 81.48 | 33.00 | 296 | 0.92 |
| | 5U70 | 92.8 | 0.271 | −5.53 | 87.54 | 39.38 | 292 | 0.91 |
| | 5TJI | 92.5 | 0.240 | −5.82 | 88.57 | 30.98 | 296 | 1.12 |
| | 5U76 | 92.9 | 0.191 | −6.34 | 84.17 | 26.35 | 297 | 1.17 |
| **Phyre2** | 5TJ6 | 95.7 | 0.239 | −5.72 | 61.63 | 37.36 | 265 | 1.01 |
| | 5U76 | 94.7 | 0.251 | −5.99 | 35.04 | 38.44 | 372 | 1.10 |

characterized by stereochemical parameters of a stable structure, with 94.2% of residues in the most favored region, 4.6% in the allowed region, and 1.2% in the disallowed region (Fig. 4). Finally, according to the QMEANBrane tool estimation, the 3D-GiK model is in the range expected for a membrane protein (Fig. 5). Figure 6 shows the monomeric and tetrameric form, and the pore cavity.

## Molecular docking

Molecular docking permits prediction of the most probable position, orientation, and conformation of interactions between a ligand and macromolecule (Ferreira et al., 2015). To predict binding free energy to GiK, 290 potassium blockers were investigated (Table S2). The overall docking energy of a given ligand molecule was expressed in kcal/mol. This approach revealed 110 molecules exhibiting the best binding free energies (−4 to −11 kcal/mol) (Table 5). Of these, 39 are commercially available compounds. Interestingly, these drugs bind in three specific hydrophobic pockets of GiK. We labeled these regions I, II, and III (Fig. 7). As shown in Table 6, 13 residues are important for binding in region I, located on the S6 transmembrane region of the channel. Of these, 10 are hydrophobic and three are polar. For region II, nine residues located on the S5–S6 linker and S6 portion of the channel interact with the various docked ligands. Of these, five are hydrophobic and four are polar. For region III, 12 extracellular residues are important for ligand interaction. Eight are hydrophobic and four are polar. The major residues observed to interact with more of the ligands were Leu65, Gly113, Gln116, Leu117, Tyr120, Met122, Phe125, Ile127, and Arg129, in region II. More negative free binding energy results in the formation of stronger complexes. We analyzed the interaction maps of the three molecules with highest binding free energies that bind to different pockets of the GiK protein. The ligand with the highest score was the $K^+$ channel blocker 6,10-diaza-3(1,3)8,(1,4)-dibenzena-1,5(1,4)-diquinolinacy clodecaphane

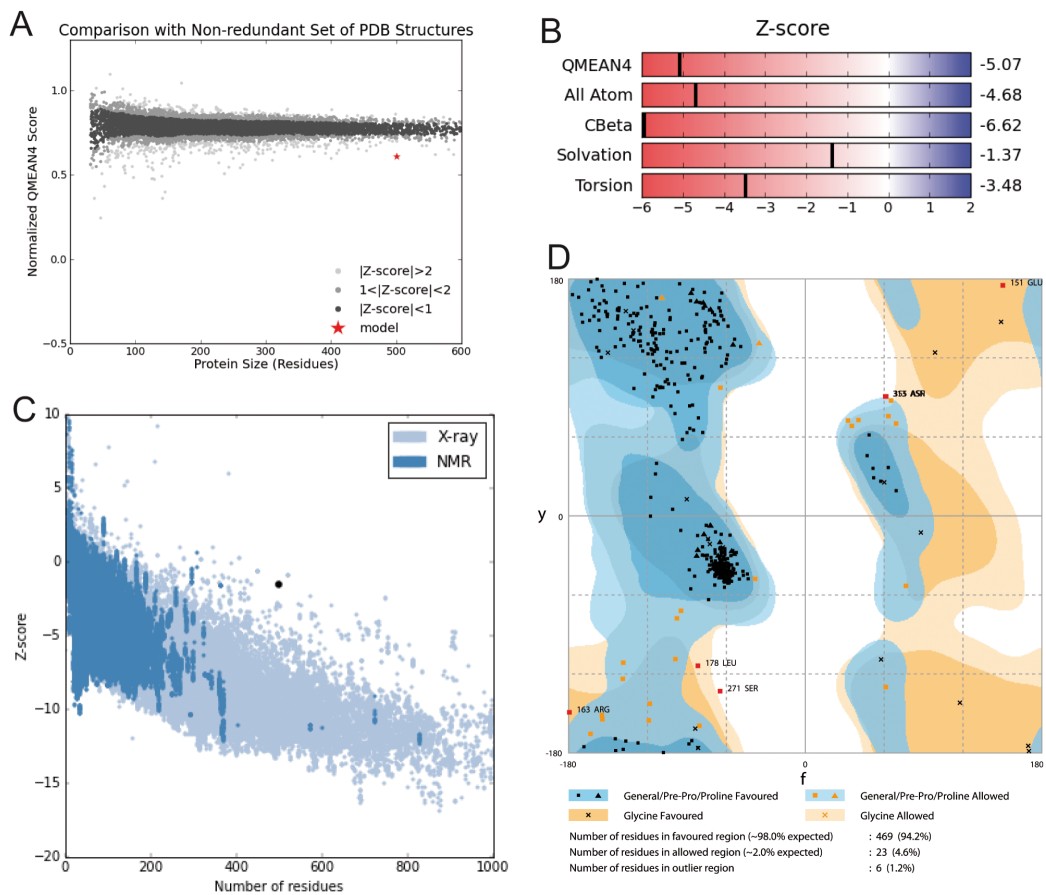

**Figure 4 Structural validation.** (A) Normalized QMEAN score of theoretical 3D structure for GiK protein model created with SWISS-MODEL server. (B) Graphical representation of the Z-Score of the individual component of QMEAN. (C) ProSA-web Z-scores of all proteins chains in PDB determined by X-ray crystallography (light blue) or NMR spectroscopy (dark blue). The Z score of GiK is highlighted as a black dot. (D) Ramachandran plot analysis, 94.2% of total residues are in the most favored region.

(UCL 1684, −11.2 kcal/mol). This drug was observed to interact with GiK in region I forming hydrophobic interactions with Phe218, Val221, Val222, Leu225, Tyr226, Val247, Leu250, and Leu276. The competitive antagonist of GABA$_A$ receptors, bicuculline, had the highest score (−10 kcal/mol) for interaction with GiK in region II. This drug forms: hydrophobic interactions with Gly113, Gln116, Leu117, Tyr118, Tyr120, Met122, Ser124, Phe125, Ser126, and Arg129. Further, bicuculline forms π-π interactions with Phe125 and Tyr68. Finally, the bioactive alkaloid, verruculogen, interacts with GiK site III by hydrophobic interactions with Val348, Pro347, Val377, Met378, and Ile411. Further, verruculogen interacts by polar interaction with Ser346 (Table 6; Fig. 8).

## DISCUSSION

In this report, we provide *in silico* evidence indicating the protein XP_001709490 from *Giardia lamblia* (GiK) is a membrane protein, with conserved potassium channels features. GiK presents seven HTM regions and the pore signature sequence SIASIGYGD.

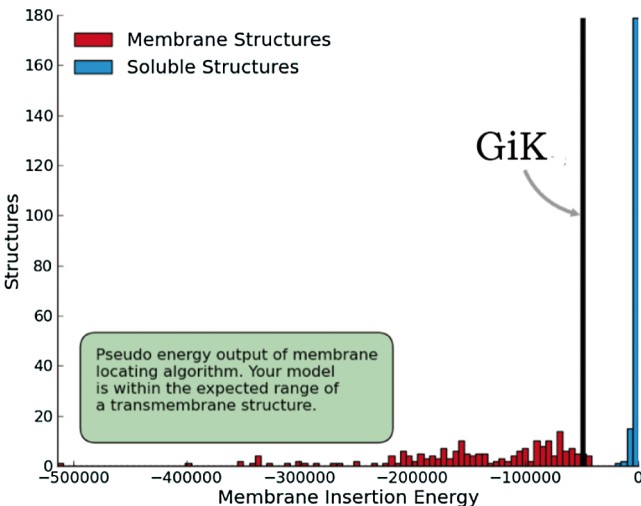

**Figure 5 Quality estimation of GiK as a membrane protein.** Prediction done with SWISS-MODEL-QMEANBrane tool.

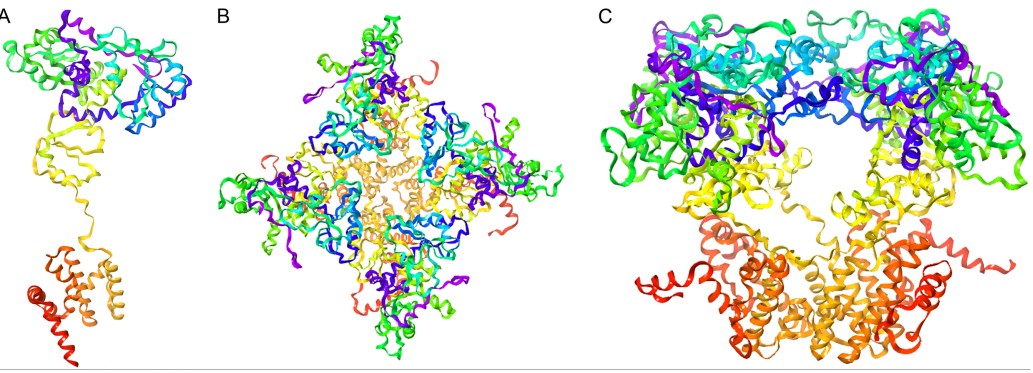

**Figure 6 Representation of the 3D-GiK modeled structure.** (A) Monomer, (B and C) Tetramer. The images were generated using Maestro software.

This sequence is associated with K$^+$ selectivity in SKCa. The presence of the Ion_trans_2 domain related to voltage-gated potassium channels suggests that GiK could be activated by either electrical means or by increasing calcium concentrations in the cell. Additional studies are necessary to understand the voltage-gated and ion selectivity in GiK.

Transmembrane protein GiK presents hydrophobic regions containing a high fraction of non-polar amino acids. It also presents hydrophilic regions containing a high fraction of polar amino acids (Figs. S1 and S4). The GRAVY value of −0.053 indicates that GiK could establish interactions with water; it can be highly hydrated in aqueous media. GiK contains protein regions that do not fold into defined tertiary structure. These are structural disorders commonly labeled IDRs. IDRs perform a central role in regulation of signaling pathways and crucial cellular processes. They are frequently associated with disease. For these reasons, there is growing interest in IDRs as potential targets for drug design (*Calcada, Korsak & Kozyreva, 2015*; *Cheng et al., 2006*).

**Table 5  Best docking score values (kcal/mol) from the potassium channel blockers to 3D-GiK model.**

| Compound | Docking score (kcal/mol) | Compound | Docking score (kcal/mol) | Compound | Docking score (kcal/mol) |
|---|---|---|---|---|---|
| UCL_1684 | −11.2 | ZINC13489790 | −8 | Flecainide | −6.9 |
| ZINC38144725 | −10.8 | Imipramine | −7.9 | Mepivacaine | −6.9 |
| Terfenadine | −10.6 | Trifluoroperazine | −7.9 | ZINC13489786 | −6.8 |
| ZINC00018512 | −10.4 | ZINC13489791 | −7.9 | ZINC13760202 | −6.8 |
| ZINC00598948 | −10.1 | ZINC13489800 | −7.9 | ZINC13777065 | −6.8 |
| Bicuculine | −10 | ZINC13489804 | −7.9 | 1-Ethyl-2-Benzimidazolinone | −6.7 |
| Cromoglicic acid | −10 | ZINC13489830 | −7.9 | ZINC13760207 | −6.7 |
| Penitrem_A | −10 | ZINC13760212 | −7.9 | ZINC13760214 | −6.7 |
| BMS_204352 | −9.4 | Linopirdine | −7.8 | ZINC03935230 | −6.5 |
| NS1643 | −9.1 | ZINC13442157 | −7.8 | ZINC13557606 | −6.5 |
| Paxilline | −9.1 | ZINC13489810 | −7.8 | ZINC13777062 | −6.5 |
| CP_339818 | −9 | ZINC13489818 | −7.8 | ZINC27617403 | −6.5 |
| Tubocurarine | −8.9 | ZINC13489829 | −7.8 | Dofetilide | −6.4 |
| ZINC13489797 | −8.8 | ZINC13489785 | −7.7 | Retigabine | −6.4 |
| UK_78282 | −8.7 | TRAM_34 | −7.6 | ZINC00005768 | −6.4 |
| Verruculogen | −8.7 | ZINC13489794 | −7.6 | ZINC13760203 | −6.4 |
| ZINC13489806 | −8.6 | ZINC13489798 | −7.6 | ZINC13777063 | −6.4 |
| ZINC13644028 | −8.6 | ZINC13489784 | −7.5 | ZINC13777067 | −6.4 |
| DIDS | −8.5 | ZINC13489803 | −7.5 | Correolide | −6.3 |
| ZINC01535217 | −8.5 | ZINC13489813 | −7.5 | ZINC03935234 | −6.3 |
| ZINC13442159 | −8.5 | ZINC13557604 | −7.5 | ZINC03935235 | −6.3 |
| ZINC38144724 | −8.5 | Amitriptyline | −7.4 | ZINC03946466 | −6.3 |
| Bicuculine methiodide | −8.4 | Dequalinium | −7.4 | ZINC13777069 | −6.3 |
| ZINC13489814 | −8.4 | ZINC01539875 | −7.4 | ZINC13777072 | −6.3 |
| ZINC13489817 | −8.4 | ZINC13489789 | −7.4 | Procaine | −6.2 |
| ZINC00015850 | −8.3 | Quinidine | −7.3 | Zoxazolamine | −6.1 |
| ZINC00603820 | −8.3 | ZINC00014006 | −7.3 | ZINC13777058 | −6 |
| ZINC01539867 | −8.2 | ZINC01535218 | −7.3 | ZINC18096411 | −6 |
| ZINC13489795 | −8.2 | ZINC13760206 | −7.3 | ZINC13777075 | −5.8 |
| ZINC13489796 | −8.2 | ZINC27617400 | −7.3 | ZINC13643922 | −5.7 |
| ZINC13489807 | −8.2 | Psora_4 | −7.2 | Chlorzoxazone | −5.5 |
| ZINC13489823 | −8.2 | ZINC18189761 | −7.2 | ZINC13579814 | −5.5 |
| ZINC29309163 | −8.2 | Pimaric_acid | −7.1 | LY_97241 | −5 |
| Niguldipine | −8.1 | Miconazole | −7 | Clofilium | −4.8 |
| ZINC13489799 | −8.1 | ZINC13760204 | −7 | Halothane | −4.5 |
| XE991 | −8 | ZINC13760205 | −7 | 4_Aminopyridine | −4.4 |
| ZINC01539870 | −8 | ZINC13760213 | −7 | | |

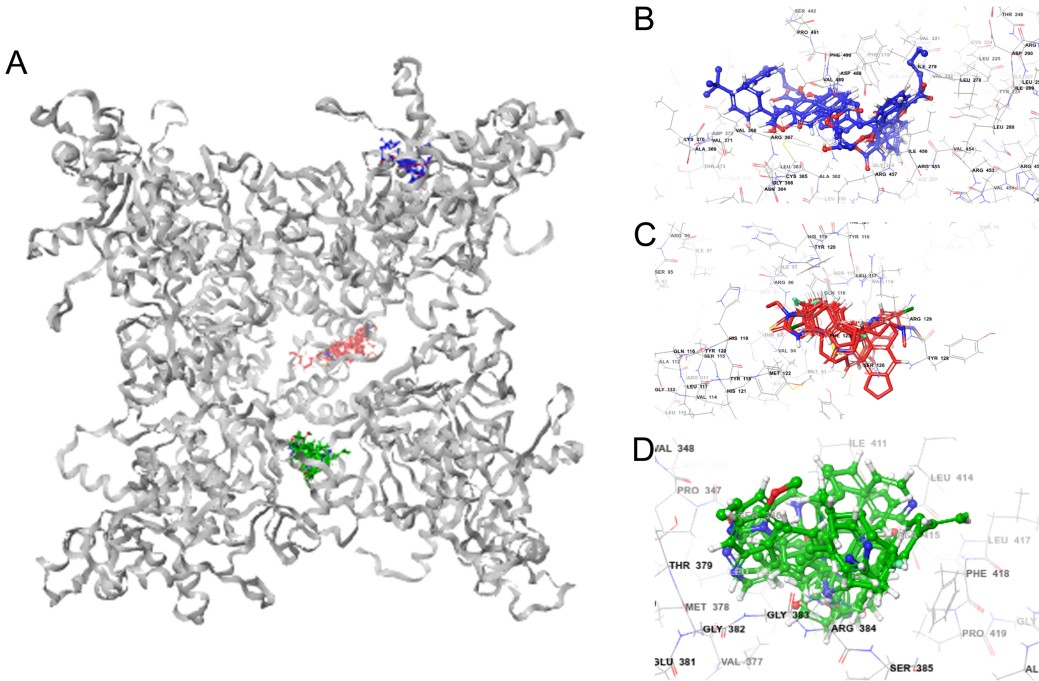

**Figure 7** GiK—potassium channel blockers docking simulations (A); (B–D) magnified views of the boxed regions depict the three potassium blockers channels binding sites (blue region I, red region II and green region III).

**Table 6** Binding sites from the potassium channel blockers to GiK.

| Region | Amino acid residues | Potassium channel blockers |
|---|---|---|
| I | Phe218, Val221, Val222, Leu225, Tyr226, Leu250, Leu278, Ile279, Ile456, Arg457, Asp488, Val489, Phe490 | UCL_1684, terfenadine, cromoglicic acid, CP_339818, niguldipine, imipramine, Psora_4, mepivacaine, procaine, chlorzoxazone, 4_Aminopyridine |
| II | Leu65, Gly113, Gln116, Leu117, Tyr120, Met122, Phe125, Ile127, Arg129 | Bicuculine, Penitrem_A, BMS_204352, NS1643, paxilline, tubocurarine, UK_78282, DIDS, bicuculine methiodide, trifluoroperazine, amitriptyline, dequalinium, miconazole, flecainide, 1-Ethyl-2-Benzimidazolinone, correolide, clofilium, halothane |
| III | Val344, Leu345, Ser346, Val377, Thr379, Gly383, Arg384, Leu388, Leu414, Ala415, Phe418, Pro419 | Verruculogen, XE991, linopirdine, TRAM_34, quinidine, pimaric_acid, dofetilide, retigabine, zoxazolamine, LY_97241 |

The prediction of 14 flexible disordered regions in GiK suggests that this protein may be important in various *Giardia* functions. This important preliminary evidence indicates that GiK is a promising subject for future study.

Potassium channels regulate multiple cellular functions in both electrically excitable and non-excitable cells. Therefore, they are attractive targets for drug design. Current trends in drug discovery focus on target identification and *in silico* compound

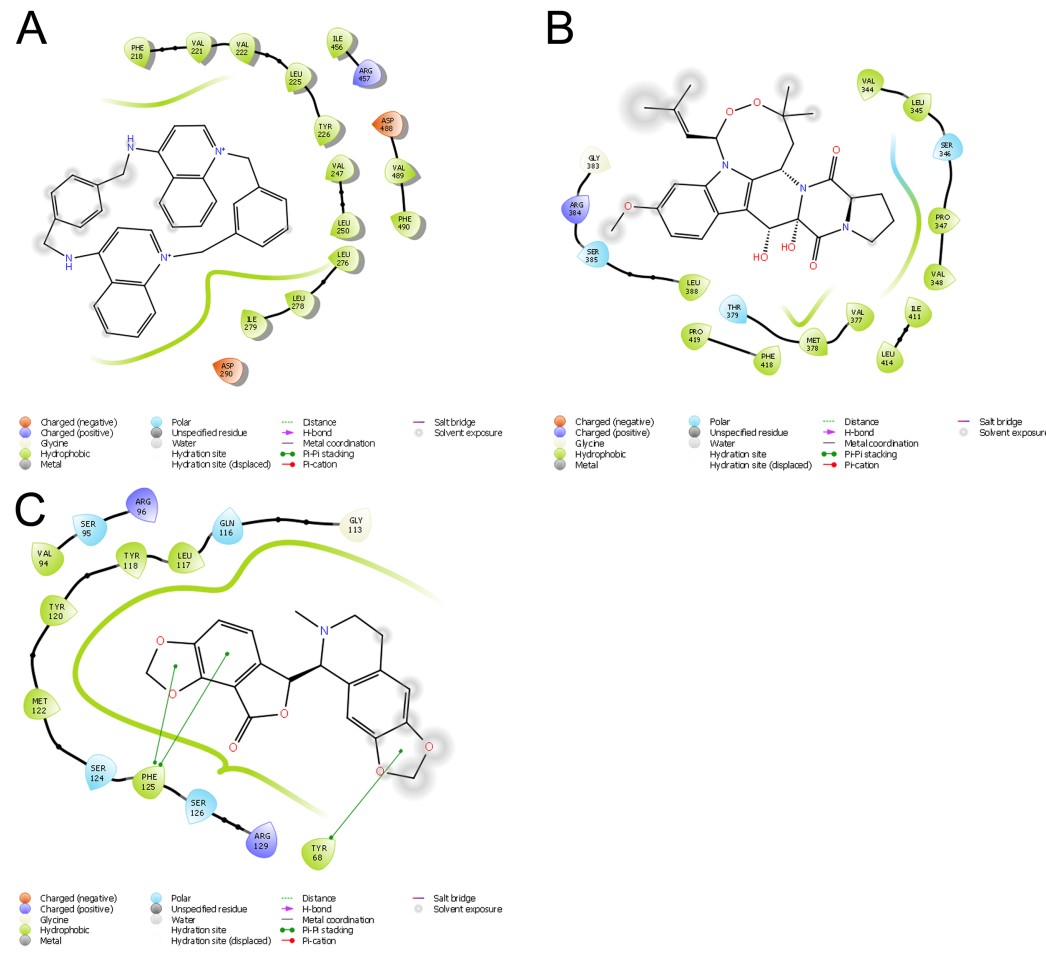

**Figure 8 Ligand interaction diagrams.** UCL 1684 (A), Bicuculline (B) or verruculogen (C). Hydrophobic interactions are depicted by green curves, π–π interactions are in green-dashed lines, and the polar interactions by curve blue lines.

design. We sought to determine whether GiK could be a potential drug target in *Giardia*. First, we built structural models of the transmembrane helical regions of GiK by homology modeling. The search for templates showed only two resolved structures: a high conductance calcium-activated potassium channel from *A. californica* (PDB ID: 5TJ6 and 5TJI) and a potassium channel subfamily T member one from *Gallus gallus* (PDB: 5U70 and 5U76). In this work, GiK presents 23% and 19% of sequence identity with the templates. RMSD is a quantitative measure of the similarity between two superimposed atomic coordinates. When using RMSD to compare protein structures, the RMSD distribution depends on the size of the protein and of the homology between the templates, among others (*Kufareva & Abagyan, 2012*). Using multiple approaches we generated 13 structural models of GiK, the quality analysis of individual models showed that even though, models obtained with Swiss model and Phyre2 had the lower RMSD values, only 50–70% of residues were modeled. The percentage of residues in the allowed regions was expected to be more than 90% for a good model. The Modeller program produced acceptable models. The best result was obtained employing the PDB ID:

5TJI (closed state); 500 aa aligned, results from a Ramachandran plot showed 94.2% of residues in the most favored region. Even though the structures obtained with 5U70 and 5TJ6 showed 90% of residues in the most favored region, the overall quality factor (*Bagal et al., 2013*) value of 5TJI is the highest (69.24%) and is within the accepted range. Besides, it is important to emphasize that in addition to RMSD, the generation of Z-score is also a measure of statistical significance between matched structures and reflects the degree of modeling success (*Dalton & Jackson, 2007*), the Z-score value (−5.07) indicates that the overall geometrical quality of the model generated by Modeller using the template 5TJI was within the acceptable range for big proteins. The overall results from RAMPAGE, QMEAN and Verify 3D indicate the 3D modeled GiK protein is of good quality. After building the 3D structure of GiK, we screened 290 potassium channel blockers. The docking results showed 110 potassium channel blockers with high affinity for the GiK protein. A total of 39 of these showed similar binding modes in three specific regions, labelled I to III. They interact principally with hydrophobic and aromatic residues such as Phe, Tyr, Leu, and Val. In agreement with results described for different potassium blockers, the ring stacking, hydrophobic interactions with several aromatic side chains and polar interactions take place mainly in S5 and S6 (*Marzian et al., 2013*; *Saxena et al., 2016*). The ionic channels can be switched or gated between an open and closed state by external signals such as changes in transmembrane voltage, binding of ligands, and mechanical stress. Some $K^+$ channels possess a highly hydrophobic inner pore that can function as an effective barrier to ion permeation (*Aryal, Sansom & Tucker, 2015*). Our results suggest that GiK is a calcium potassium activated channel with a hydrophobic inner pore. Additional research is needed to confirm this finding. We plan to expand our studies in this area in the future (*Liu & Kokubo, 2017*; *Martins et al., 2018*).

Other authors have reported successful computational screening of $K^+$ channels. These reports demonstrate that computational screening is an effective method for rapidly discovering new channels blockers from large databases (*Kingsley et al., 2017*; *Liu et al., 2003*). Hong Liu and coworkers identified 14 natural compound of relatively lower binding energy. These researchers used a docking virtual screening approach based upon a 3D model of the eukaryotic $K^+$ channels. Experimental results showed that four of these exerted potent and selective inhibitory effect on $K^+$ channels (*Liu et al., 2003*). Interestingly, some of the potassium channel blockers in our study have been employed with some success for their antiparasite activity. Verruculogen, clofilium, clotrimazole, trifluoroperazine, bicuculline methiodide, tubocurarin, and dequalinium chloride affect the growth of *Trypanosoma bruceii, Leishmania donovani, Plasmodium falciparum,* and *Trichomonas vaginalis* (*Della Casa et al., 2002*; *Nam et al., 2011*; *Rateb et al., 2013*; *Waller et al., 2008*). Quinidine inhibits the cell division in *Tetrahymena pyriformis* (*Conklin, Heu & Chou, 1970*). Trifluoperazine alters the motility in *Paramecium sp.* (*Otter, Satir & Satir, 1984*). Disodium cromoglycate and terfenadine show activity in infection models of *Toxoplasma gondii* and *Plasmodium yoelli nigieriensis* (*Rezaei et al., 2016*; *Singh & Puri, 1998*). In *Giardia lamblia*, trifluoroperazine, a calmodulin

antagonist, inhibits excystment (*Bernal et al., 1998*). It remains uncertain whether potassium channels are the targets of these compounds.

## CONCLUSION

Using structural bioinformatics, we identified the hypothetical protein XP_001709490 from *Giardia lamblia* as a potassium channel, GiK. By protein docking analysis, we found 39 commercial potassium channel blockers that have affinity for this protein. These blockers are predicted to bind in three specific regions on the protein. The novelty of this work lies in the use of the model 3D-GiK structure to screen compounds with theoretical affinity. Some of the drugs predicted by the model to be effective have demonstrated antiparasitic activity in *in vitro* and *in vivo* assays. Experimental analyses are needed to confirm the activity of these drugs on *Giardia*. The low homology of GiK with proteins in the human genome contributes to its potential as a target of specific pharmacological agents.

### Funding

Lissethe Palomo, Filiberto Gutiérrez and Verónica Ochoa awarded scholarships 377019, 574252 and 575532 from CONACYT. This work was partially supported by the Fondo Sectorial de Investigación en Salud y Seguridad Social (CONACYT-FOSISS 2015-1-261442). There was no additional external funding received for this study. The funders had no role in study design, data collection and analysis, decision to publish, or preparation of the manuscript.

### Grant Disclosures

The following grant information was disclosed by the authors:
Lissethe Palomo, Filiberto Gutiérrez and Verónica Ochoa awarded scholarships: 377019, 574252 and 575532 from CONACYT.
Fondo Sectorial de Investigación en Salud y Seguridad Social: CONACYT-FOSISS 2015-1-261442.

### Competing Interests

The authors declare that they have no competing interests.

### Author Contributions

- Lissethe Palomo-Ligas conceived and designed the experiments, performed the experiments, analyzed the data, prepared figures and/or tables, authored or reviewed drafts of the paper, approved the final draft.
- Filiberto Gutiérrez-Gutiérrez conceived and designed the experiments, performed the experiments, analyzed the data, contributed reagents/materials/analysis tools, prepared figures and/or tables, authored or reviewed drafts of the paper, approved the final draft.
- Verónica Yadira Ochoa-Maganda performed the experiments, analyzed the data, authored or reviewed drafts of the paper, approved the final draft.

- Rafael Cortés-Zárate contributed reagents/materials/analysis tools, authored or reviewed drafts of the paper, approved the final draft.
- Claudia Lisette Charles-Niño contributed reagents/materials/analysis tools, authored or reviewed drafts of the paper, approved the final draft.
- Araceli Castillo-Romero conceived and designed the experiments, analyzed the data, contributed reagents/materials/analysis tools, prepared figures and/or tables, authored or reviewed drafts of the paper, approved the final draft.

### Data Availability

Raw data are available in the Supplemental Files.

### Supplemental Information

Supplemental information for this article can be found online at http://dx.doi.org/10.7717/peerj.6430#supplemental-information.

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
