# Peer review of "Identification of a novel potassium channel (GiK) as a potential drug target in Giardia lamblia: Computational descriptions of binding sites"

_PeerJ, doi:10.7717/peerj.6430_

## Round 0.1 · original submission · Major Revisions

Dear Authors,

Based on the reviewers' reports and my own reading, your manuscript requires major revision. When you submit your revised manuscript, please address all the reviewers' comments in detail. That will greatly speed up the re-review.

Reviewer 1 ·

Basic reporting

There are abundant grammatical errors, and English revision is required. See examples below, with many others throughout the text.
52: “The ionic” channels should be changed to simply “ionic channels”
70: “Others, emphasizes…”
72: “In another hand…”
163: “…we found that GiK has a 34.6% relation with potassium channels, and its taxonomy is related to bacteria in a 63%.”

Many phrases need references, i.e. lines 47-50.

Experimental design

The present work by Palomo-Ligas et al. describes the generation of a homology model of and screening of potential ligands of a potential potassium channel, which is a potential target for drugs against Giardia lamblia infection. The research appears to be timely and of sufficient importance, however, the manuscript is very terse (results is <50 lines), and should be expanded to include further comparisons to experimental data of similar ion channels. Further, the impact of the work would be higher if the authors evaluate the binding thermodynamics and stability of the best ranked poses (e.g. as in Martins et al. JCAMD, 2018, 32: 591-605), for instance by MM/GBSA or MM/PBSA, and Molecular Dynamics of the channel embedded in a membrane with different ligands, possibly followed by free energy calculations.

Validity of the findings

Introduction:
76-77: “However, the structural characterization of ionic channels in this 

protozoan and its possible role as drug targets is poorly understood.” Should be elaborated to mention what is known about them, or similar receptors.

Results:
148: “showed 24-50% identity and 50-90% positivity”. Positivity or similarity?

The receptor has a large number (>1400) of residues, and it is not clear if the entire protein, or a specific domain was modeled. Alignments of domains and sequences suggest that only the transmembrane section was included, which, depending on what is known about the importance of other segments, may be acceptable. Figure 1 should be updated to highlight the relevant section, and perhaps reduce the representation of the region not included in the model.

A disorder prediction of the sequence highlighting the modeled region should be included. If it includes disordered segments, the modeling approached used, which is unlikely to handle such regions well, must be justified.

Please describe how the tetrameric form was obtained in Figure 4.

Was a specific region of the channel specified in the docking, or the whole receptor? This needs to be clarified.

The docking section needs to be elaborated, for instance, to include a visual comparison of the binding modes found to experimental data of ligand binding to similar receptors. Do the poses localize to known pockets? What affinities are expected for the different binding pockets (based on experimental data for similar channels in the literature)? Which pocket is typically targeted in these channels?

Discussion:
Do the most promising ligands identified have similar functional groups, properties, etc as those known to bind to similar to channels? A figure showing binding site and molecule similarity to homologous K channels would be helpful.

Reviewer 2 ·

Basic reporting

The Introduction provides sufficient background for the readers to understand the research problem. It will be better if the authors can briefly describe the molecular basis of the standard therapy (i.e. nitroimidazole compounds) of giardiasis, and how targeting potassium channels is a novel/different therapeutic approach.

The most important issue, however, is the lack of details in the Materials & Methods, and the Results sections make it hard if not impossible for the future readers to reproduce the analyses described in this work.

Experimental design

In this work, the authors used protein sequence analyses and molecular modelling to construct a three-dimensional model of a putative Giardia potassium channel GiK. Based on this structural model, the authors then performed docking virtual screening to search for potential binding sites of potassium channel-blockers.

The Materials and Methods section needs a lot more details. For instance, the authors used multiple web tools to perform sequence analyses of GiK, however, the parameters used in these analyses were not described. I suggest to include the details of how these analyses were setup in the supplemental materials so that readers who are interested in can reproduce the analyses. Also, the authors should state what information they intended to obtain from each web tool.

In Line 94, the authors mentioned that 51 potassium channel genomic sequences of different species were used to identify potential K channel in Giardia lamblia. The authors should list all 51 sequences used.

Validity of the findings

The results from bioinformatic sequence analysis, molecular modeling, and docking were summarized in the Results section, however, more detail analyses should be provided.

1. In Figure 2, the authors should clearly state the source of each result presented. For instance, ion_trans_2 is the result from NCBI Conserved Domains (?).

2. In Line 163-164, the authors mentioned that "......by the Protein Data Bank
platform, we found that GiK has a 34.6% relation with potassium channels, and its
taxonomy is related to bacteria in a 63%". It is unclear how these two numbers were obtained from PDB.

3. Based on the UniProtKB accession number of GiK, it seems that the protein has 1416 amino acids. The author should state clearly which domain(s) or region of the protein they modelled.

4. The authors used multiple modelling programs to construct the three-dimensional models of (part of ) GiK. They then used different web tools to verify the quality of their models. Even though multiple approaches have been used to generate 13 structural models (listed in Table 4) of GiK, the authors only showed one of them in Figure 4. In order to demonstrate the reliability of this modelling approach, the authors should perform a detail comparison of the structural models they obtained from different programs and templates. One would expect that if this approach is robust, the models obtained should be similar.

---

## Round 0.2 · Major Revisions

As the reviewers point out, some of the validation is still lacking and should be provided.

Reviewer 1 ·

Basic reporting

General remarks:
-The revised manuscript addresses most of the concerns associated with the first version
-It should be suitable for publication once the items below have been addressed
-Another round of revision is not required

Issues:
-Figure 2 is of low resolution
-Labels for S5, S6, etc should be added to figure 7
-Line 72: Remove comma after "Following"
-Line 73: Use micro symbol, µ, and add space between value and units
-Line 242: Ley117
-Line 245: Remove extra space (Ile 411)

Experimental design

no comment

Validity of the findings

no comment

Additional comments

Even if the additional analyses of pose stability and free energy by molecular dynamics simulations are to be left for a future study, it would be worthwhile to mention in the discussion section that this is a logical next step before experimental investigations, accompanied by a reference or two: (1) Martins et al. JCAMD, 2018, 32: 591-605 and (2) Kokubo et al. J. Chem. Inf. Model., 2017, 57: 2514-2522.

Reviewer 2 ·

Basic reporting

Even though the authors mentioned that the manuscript was edited for correct English language, there are a large number of grammatical errors in the revised version,

Experimental design

The authors have included more details in the Materials and Methods section. Unfortunately, it is still not clear how the tetrameric form of GiK was generated. Did the authors generated it based on the monomeric form? If that is the case, details on how this was done should be included.

Validity of the findings

The authors have used multiple approaches to generate 13 structural models of GiK, In Table 4, they reported the quality of individual models assessed by different web tools. However, there is no comparison between the models. For instance, what are the RMSD between the models?

Additional comments

The authors have revised the manuscript based on some of the reviewers' suggestions/questions. Unfortunately, I found that some of the questions were still not properly addressed. In addition, there are many grammatical errors in the manuscript.

---

## Round 0.3 · accepted · Accept

You have made the requested changes. I am pleased to accepted your manuscript.

#